# Identification and Characterization of a New Cold-Adapted and Alkaline Alginate Lyase TsAly7A from *Thalassomonas* sp. LD5 Produces Alginate Oligosaccharides with High Degree of Polymerization

**DOI:** 10.3390/md21010006

**Published:** 2022-12-22

**Authors:** Chengying Yin, Jiaxia Sun, Hainan Wang, Wengong Yu, Feng Han

**Affiliations:** 1Laboratory for Marine Drugs and Bioproducts of Qingdao Pilot National Laboratory for Marine Science and Technology, Qingdao 266237, China; 2Key Laboratory of Marine Drugs, Ministry of Education, Ocean University of China, Qingdao 266003, China; 3Shandong Provincial Key Laboratory of Glycoscience and Glycoengineering, Ocean University of China, Qingdao 266003, China; 4School of Medicine and Pharmacy, Ocean University of China, Qingdao 266003, China

**Keywords:** alginate lyase, endo-type, alkaliphilic, polyM-preferred, cold-adaption, high degree of polymerization

## Abstract

Alginate oligosaccharides (AOS) and their derivatives become popular due to their favorable biological activity, and the key to producing functional AOS is to find efficient alginate lyases. This study showed one alginate lyase TsAly7A found in *Thalassomonas* sp. LD5, which was predicted to have excellent industrial properties. Bioinformatics analysis and enzymatic properties of recombinant TsAly7A (rTsAly7A) were investigated. TsAly7A belonged to the fifth subfamily of polysaccharide lyase family 7 (PL7). The optimal temperature and pH of rTsAly7A was 30 °C and 9.1 in Glycine-NaOH buffer, respectively. The pH stability of rTsAly7A under alkaline conditions was pretty good and it can remain at above 90% of the initial activity at pH 8.9 in Glycine-NaOH buffer for 12 h. In the presence of 100 mM NaCl, rTsAly7A showed the highest activity, while in the absence of NaCl, 50% of the highest activity was observed. The rTsAly7A was an endo-type alginate lyase, and its end-products of alginate degradation were unsaturated oligosaccharides (degree of polymerization 2–6). Collectively, the rTsAly7A may be a good industrial production tool for producing AOS with high degree of polymerization.

## 1. Introduction

Alginate is a natural linear anionic polymer which consists of β-D-mannuronic acid (M) and α-L-guluronic acid (G) linked by β-1,4-glycosidic bonds [1]. It is the only natural marine biological polysaccharide with one carboxyl in each sugar ring [2]. Alginate polymer blocks arrange in three possible ways: poly-α-L-guluronic acid (polyG), poly-β-D-mannuronic acid (polyM), and hetero-polymeric random sequences (polyMG). These characteristics lead to the difference of their high-order structures, so polyM, polyG and their derivatives display different activities [3]. In medical fields, alginates with different arrangement and degree of polymerization (DP) have wide application prospects, including drug delivery and tissue engineering [4,5,6]. Alginate oligosaccharides (AOS) and their derivatives with different DPs are becoming popular due to their favorable biological activity and water solubility [1]. Most importantly, AOS have been found to play an important role in anti-tumor [7], anti-inflammatory [8], neuroprotective [9], immune regulation [10], anti-obesity [11], antibacterial [12], antioxidant [13,14], anti-diabetic [15] and other aspects [16,17]. These functions of AOS were mostly relevant to gut microbiota. For example, Zhang et al. proposed that fecal microbiota transplantation (FMT) from AOS-dosed mice improved small intestine function by increasing beneficial microbes [8], and another study showed that GV-971 could suppress neuroinflammation through inhibiting gut dysbiosis to reduce phenylalanine/isoleucine accumulation [9]. In addition, Li et al. determined that unsaturated alginate oligosaccharides (UAOS) obtained by enzyme degradation showed significant anti-obesity effects in a high-fat diet (HFD) mouse model [18], and then they determined that UAOS can attenuate the HFD-induced obesity through modulating gut microbiota by selectively increasing the relative abundance of beneficial intestinal bacteria and decreasing the abundance of inflammogenic bacteria [19]. The different types of AOS had different functions in past studies, including the immuno-stimulatory activity of guluronate oligosaccharide (GOS) [10], the hypoglycaemic and hypolipidaemic activities of oligosaccharide from *S. confusum* (SCO) [15] and the neuroprotective activity of GV-971 (a sodium oligomannate) as mentioned above [9]. It is worth mentioning that UAOS performed significant anti-obesity effects compared with saturated alginate oligosaccharides (SAOS) [18,19].

At present, enzymatic degradation is the most common method to prepare AOS, so the key to producing functional AOS is to find efficient alginate lyases [16,17]. According to amino acid sequence, alginate lyases are divided into 12 polysaccharide lyase families (PL5, 6, 7, 14, 15, 17, 18, 31, 32, 34, 36, 39, 41) in the CAZy database [20]. The PL7 family (http://www.cazy.org/PL7.html, accessed on 11 February 2022) contains the most alginate lyases and is further divided into six subfamilies [21,22]. In addition, based on substrate specificity, it can be classified into polyG-specific, polyM-specific, bifunctional alginate lyase and polyMG-specific alginate lyase [23,24,25]. Based on the different modes of action, it can also be categorized into endo- and exo-type alginate lyases [26]. Endo-type alginate lyases can cleave the glycosidic bonds in alginate polymer randomly and release unsaturated oligosaccharides (disaccharides, trisaccharides and tetrasaccharides) as the main products. Exo-type alginate lyases cut the alginate chains successively from non-reducing ends to produce monosaccharides.

This study cloned and expressed a new PL7 alginate lyase-encoding gene, *tsaly7A*, from *Thalassomonas* sp. LD5. The recombinant TsAly7A (rTsAly7A) exhibited good properties such as pH stability under alkaline conditions, high activity under low temperature, and wide range of product distribution. These characteristics make rTsAly7A a good industrial production tool for producing AOS with high DPs.

## 2. Results

### 2.1. Sequence Analysis of TsAly7A

One predicted alginate lyase gene, *tsaly7A*, was detected and cloned from *Thalassomonas* sp. LD5, composed of 939 bp, encoding 312 amino acid residues. It only contained a catalytic module (CM) as shown in Figure 1A. The original length of *tsaly7A* (*OL-tsaly7A*) had a signal peptide (SP) at the N-terminal end with a length of 17 amino acid residues, a carbohydrate binding module (CBM) at middle with a length of 161 amino acid residues and a CM at C-terminal end (Figure 1A). The theoretical molecular weight of TsAly7A was 34.39 kDa and theoretical pI was 4.57. The sequence data were deposited in GenBank with accession No. OM672104.1. According to the results of Protein BLAST search, the similarity rate between TsAly7A and AlgMsp of PL7 family from *Microbulbifer* sp. 6532A [27] was 71%, indicating that TsAly7A was a new member of PL7. Further phylogenetic analysis proved that TsAly7A belonged to the fifth subfamily of PL7 (Figure 1B).

### 2.2. Expression and Purification of rTsAly7A

The rTsAly7A was successfully constructed and then expressed in *E. coli* BL21(DE3). By optimizing the induction conditions of rTsAly7A, the yield of rTsAly7A was highest at 18 °C and induced by 0.1 mM IPTG for 24 h (Figure 2A). After 1 L fermentation broth was purified, 13.46 mg pure enzyme of rTsAly7A was finally obtained. Through purification, the specific activity of rTsAly7A was 1536.36 U/mg, and the recovery rate was 31.41%. A single band on SDS-PAGE gel showed the molecular weight of the rTsAly7A was estimated to be about 40 kDa (Figure 2B).

### 2.3. Biochemical Characterization of the rTsAly7A

The optimum pH of rTsAly7A was 9.1 in Glycine-NaOH buffer (Figure 3A), while the enzyme activity at pH 7.0 was less than 50% of the highest. The enzyme activity of rTsAly7A remained above 80% after 12 h incubation in Na_2_HPO_4_-NaH_2_PO_4_ buffer (pH 7.0–8.0) and was most stable at pH 8.9 in Glycine-NaOH buffer for 12 h (Figure 3B), indicating that it was alkaliphilic. 

The optimum temperature of rTsAly7A was 30 °C, but the enzyme activity decreased sharply over 30 °C (Figure 3C), whereas it exhibited 16% of highest activity at 0 °C. In addition, after incubation for an hour at 20 °C, it maintained half of the enzyme activity (Figure 3D). Therefore, rTsAly7A is a cold-adapted alginate lyase that can be used at room temperature.

Only Fe^3+^ promoted the enzyme activity by 1.5 times as shown in Figure 4A. The enzyme activity of rTsAly7A was significantly decreased with 1 mM Li^+^, Cu^2+^, Co^2+^, Ba^2+^, Ca^2+^ and Ni^2+^ (Figure 4A). As for 1mM Zn^2+^, the enzyme activity of rTsAly7A was mostly lost. In the presence of EDTA, the enzyme activity of rTsAly7A was even completely lost. In addition, it is noteworthy that 1mM Na^+^, K^+^, NH_4_^+^, Mg^2+^, Mn^2+^, Fe^2+^, and SDS had no significant effect on rTsAly7A. As shown in Figure 4B, rTsAly7A maintained 50% activity in the absence of NaCl and reached maximum activity in the presence of 100 mM NaCl.

### 2.4. The Substrate Specificity of rTsAly7A

The 0.3% (*w/v*) substrate was prepared under the optimum pH 9.1 and NaCl (100 mM) conditions, and the enzyme activity was determined by using substrate alginate, polyM and polyG, respectively. The degradation ability of polyM was 76% of that of alginate, and the degradation ability of polyG was weak, only 12% of that of alginate (Figure 4C).

### 2.5. Degradation Mode and End-Products of rTsAly7A

Size-Exclusion Chromatography (SEC) was used to reveal the time-course of alginate degradation by rTsAly7A. At the beginning of the degradation reaction, a large number of products with high DP appeared (Figure 5A). With the extension of degradation time, these products gradually degraded into oligosaccharides with lower DPs. It indicated that rTsAly7A was an endo-type alginate lyase, and some products with low DPs appeared at the initial stage of degradation reaction, indicating that the initial enzymatic reaction speed of the enzyme was very fast.

The SEC results of the final degradation products of rTsAly7A showed five UV absorption peaks at 12.6 mL, 13.2 mL, 13.9 mL, 14.7 mL and 15.6 mL, respectively, with a ratio of 0.24:1.07:1.78:1.16:1 (Figure 5B). The five peaks were collected and analyzed by ESI-MS. The results of mass spectrometry analysis were shown in Figure 5C. There are several obvious nuclear-to-mass ratio peaks in the mass spectrometry results, 351.06, 527.09, 703.12, 879.15 and 1055.18 *m/z* representing molecular peaks [∆DP2−H]^−^, [∆DP3−H]^−^, [∆DP4−H]^−^, [∆DP5−H]^−^, and [∆DP6−H]^−^, respectively, which correspond to the molecular weights of unsaturated alginate disaccharide, trisaccharide, tetrasaccharide, pentasaccharide and hexasaccharide. Therefore, the final degradation product of rTsAly7A were unsaturated oligosaccharides of DP 2–6.

## 3. Discussion

In this study, we characterized an endo-acting, cold-adapted and polyM-preferred alginate lyase TsAly7A from *Thalassomonas* sp. LD5. Notably, different from our previous work on TsAly7B [28], which produced unsaturated oligosaccharides of DP 2–4 as its final products, TsAly7A released DP 2–6 from alginate. 

The results reflected that rTsAly7A had a lower optimal temperature (30 °C) and maintained 16% of highest activity at 0 °C, which indicates that rTsAly7A is one cold-adapted alginate lyase. The cold-adapted character of rTsAly7A reveals its adaptation to marine environment in that TsAly7A was cloned from marine bacterium *Thalassomonas* sp. LD5 [29], which was found in the coastal sediments with a temperature of 5 °C. Some cold-adapted alginate lyases had been characterized, but rTsAly7A had some excellent properties in other aspects. For example, AlyS02 from *Flavobacterium* sp. S02, AlyPM from *Pseudoalteromonas* sp. SM0524 and AlgSH17 from *Microbulbifer* sp. SH-1 were all cold-adapted and polyM-preferred alginate lyases [30,31,32], but AlyS02 and AlyPM could only release the oligosaccharides of DP 2, 3, and their optimal pH was 7.6 and 8.5, respectively. AlgSH17 could release the oligosaccharides of DP 2–6, but it was not really stable as rTsAly7A under alkaline conditions. TsAly7A released unsaturated oligosaccharides of DP 2–6 from alginate and had the highest activity in pH 9.1. In addition, rTsAly7A was stable under alkaline conditions as shown in Figure 3B. 

The alkali suitability is one good property in alginate lyase application and alginate oligosaccharide production. Several studies have reported some robust alginate lyase; Alyw203 from *Vibrio* sp. W2 showed outstanding pH stability with a highest activity under alkaline conditions of pH 10.0 [29] and Aly08 from *Vibrio* sp. SY01 held above 80% of its original activity in pH 4.0–10.0 [33]. Similarly, the rTsAly7A showed outstanding pH stability under alkaline conditions, which made it an excellent tool in strict industrial condition. 

This study also demonstrated that rTsAly7A had a wide substrate utilization range with a preference for polyM as most alginate lyase from PL7. Similarly, AlyPM and AlgSH17 also preferably degraded polyM [31,32] while AlyS02 preferably degraded polyG [30].

The study of the effect of ions has shown that only Fe^3+^ promoted enzyme activity. However, Li^+^, Cu^2+^, Co^2+^, Ba^2+^, Ca^2+^ and Ni^2+^ inhibited the enzyme activity of rTsAly7A, which means that it cannot be used with these conditions. Additionally, Zn^2+^ and EDTA showed remarkable inhibitory effects on rTsAly7A, and yet rTsAly7A remained at 89% activity in the presence of SDS, which indicates it may have more extensive use in application. In addition, Na^+^, K^+^, NH_4_^+^, Mg^2+^, Mn^2+^ and Fe^2+^ had no effect on rTsAly7A. The enzyme activity of rTsAly7A was the highest in the 100 mM NaCl condition and rTsAly7A can maintain a 50% activity without NaCl, which means it would not easily cause equipment corrosion in subsequent industrial production applications without using NaCl. 

Therefore, as mentioned above, rTsAly7A is a tool to produce high degree of polymerization oligosaccharides. Studies by Chen et al. showed that the oligosaccharides of DP 5 released by alginate lyase had a remarkable inhibitory effect on the growth of osteosarcoma cells, while DP 2, 3 and 4 had no inhibitory effect [34]. The most common products of alginate lyase were DP 2–4 [30,35,36]. In other words, the oligosaccharides of DP 2–6 released by rTsAly7A may have more new properties can be studied. 

## 4. Materials and Methods

### 4.1. Strains, Media, Plasmids, and Reagents

*Escherichia coli* strains BL21 (DE3) and DH5α from TaKaRa (Dalian, China) were cultivated in Luria–Bertani (LB) medium containing Kanamycin (50 μg/mL) when necessary. For the expression of recombinant proteins, plasmid pET-24a (+) was used. The DNA polymerase and DNA Restriction enzyme were from TaKaRa (Dalian, China). TIANamp Bacteria DNA Kit was purchased from TIANGEN BIOTECH (Beijing, China). ClonExpress II One Step Cloning Kit was purchased from Vazyme (Nanjing, China). Qingdao Gather Great Ocean Algae Industry Group Co., Ltd. (Qingdao, China) provided Alginate and polyM, polyG were from Qingdao HEHAI Biotech Co., Ltd. (Qingdao, China). 

### 4.2. Identification, Cloning and Sequence Analysis of TsAly7A

Genomic DNA used as template was extracted from *Thalassomonas* sp. LD5 using TIANamp Bacteria DNA Kit. PCR primers (TsAly7A-F: taagaaggagatatacatatgGTGGTTAATCACTGTGGTGAACTTG, TsAly7A-R: gtggtggtggtggtgctcgagATAGTTATAGCCGGTATGTGAATTGTC) were designed according to the genomic sequence of *Thalassomonas* sp. LD5 to obtain gene Tsaly7A without signal peptide and stop codon. The vector pET-24a (+) was linearized by restriction enzyme *Nde* I and *Xho* I. Then, the gene Tsaly7A was ligated into pET-24a (+) by ClonExpress II One Step Cloning Kit using primers above. The SignalP-5.0 server (http://www.cbs.dtu.dk/services/SignalP/, accessed on 9 December 2020) was used to predict Signal peptide [37]. The recombinant plasmids rTsAly7A were transformed into the *E. coli* DH5α. Theoretical molecular weight and pI were determined using the ProtParam tool (https://web.expasy.org/protparam/, accessed on 9 December 2020) [38]. Multiple sequence alignments and phylogenetic tree construction of TsAly7A were performed using MAGA-X [39,40,41,42]. 

### 4.3. Expression and Purification of rTsAly7A

According to Zhang et al. [28], protein expression was conducted in *E. coli* BL21 (DE3) strains induced until OD_600_ reached 0.6 with 0.1 mM IPTG for 24 h. The cells were collected, resuspended, and broken up, and then the crude enzymes were extracted from the supernatants according to Zhang et al. [28]. One 1 mL HisTrap^TM^ HP Column (GE healthcare, Stanford, USA) was then used to separate the recombinant proteins from the crude enzyme. SDS-PAGE on a 10% (*w/v*) resolving gel was used to detect the purity and molecular mass of recombinant TsAly7A, and the NCM BCA protein assay kit (NCM Biotech, Suzhou, China) was used to measure the protein content. 

### 4.4. Activity Assay of rTsAly7A

Alginate lyase activity was determined by UV spectrophotometry for its change at 235 nm. Briefly, 900 μL of 0.3% (*w/v*) of alginate substrate (50 mM PB, 100 mM NaCl, pH 9.1) was incubated at 30 °C for 5 min, and then 100 μL of enzyme solution was added. Enzyme boiled at 100 °C for 10 min was used as the control. The A_235_ value was detected by UH5300 UV–visible spectrophotometer (HITACHI, Tokyo, Japan) after being incubated at 30 °C for 10 min. An enzyme activity unit (U) was defined as the amount of enzyme required to increase 0.1 units of UV absorption per minute. These results were repeated 3 times and the average values were indicated along with a standard deviation. 

### 4.5. Biochemical Characterization of rTsAly7A

The enzyme activity was determined at 10–60 °C to find the optimum temperature of rTsAly7A. To determine the thermal stability, the residual activity of the enzyme was determined after incubation at 0–80 °C for 1 h. The substrate was prepared with 50 mM buffers at different pH [Na_2_HPO_4_-citric acid (pH 2.2–8.0), Na_2_HPO_4_-NaH_2_PO_4_ (pH 5.8–8.0), Tris-HCl (pH 7.1–8.9), Glycine-NaOH (pH 8.6–10.6)], and the enzyme activity was measured at the optimum temperature to study the optimum pH value of rTsAly7A. To study its pH stability, the enzyme was incubated in different pH at 0 °C for 12 h, and then its residual activity was determined. As for the effect of sodium chloride on rTsAly7A activity, 0.3% (*w/v*) alginate substrate (50 mM glycine-NaOH, pH 9.1) was prepared by adding different concentrations (0–1 M) of NaCl. To determine the effects of different metal ions and surfactants on TsAly7A, the enzyme activity of rTsAly7A was measured by adding 1 mM different metal ions and SDS at optimal temperature and pH. To determine the substrate specificity of rTsAly7A, 0.3% (*w/v*) of different substrate (polyM, polyG, and alginate) solutions were used to determine the activity of it.

### 4.6. Degradation Mode and End-Products of rTsAly7A

To clarify the degradation mode of action of rTsAly7A, 1 mL (50 U) enzyme was put in 9 mL alginate substrate [0.3% (*w/v*), 50 mM PB, 100 mM NaCl, pH 9.1] and incubated at 30 °C with progressive time (0, 1, 5, 10, 20, 30 and 60 min, respectively). The reaction was ended by boiling for 10 min. Subsequently, the degradation mode was further detected by fast protein liquid chromatography (FPLC) with a Superdex peptide 10/300 GL column (GE Healthcare, Boston, MA, USA) for separation, 200 mM NH_4_HCO_3_ at a flow rate of 0.2 mL/min was used as the mobile phase, and UV detector was used to detect A_235_. 

To obtain the final product, rTsAly7A was put in 0.3% (*w/v*) alginate solution to produce a final concentration of 100 U/mL, and then incubated for 12 h at 30 °C. The obtained sample was investigated by gel filtration on Superdex peptide 10/300 GL column. The detection wavelength was 235 nm, the flow rate of the mobile phase (0.2 M NH_4_HCO_3_) was 0.2 mL/min. In addition, each peak of the final product was collected, and then mixed with acetonitrile 1:1 (*v/v*). After that, its molecular weight was detected by negative-ion electrospray ionization-mass spectrometry (ESI-MS) in the range of 100–2000 *m/z*. 

## 5. Conclusions

In this study, an endo-acting, cold-adapted, and polyM-preferred alginate lyase rTsAly7A from *Thalassomonas* sp. LD5 was detailed. rTsAly7A had a low optimal temperature (30 °C) and remained at 16% of highest activity at 0 °C, which indicated that rTsAly7A is one cold-adapted alginate lyase. Compared with other characterized alginate lyases, its pH stability under alkaline conditions was pretty good in that it can remain at above 90% of activity after incubation at pH 8.9 in Glycine-NaOH buffer for 12 h. rTsAly7A shared the highest activity in the presence of 100 mM and maintained 50% of the highest activity in the absence of NaCl. The SEC results showed rTsAly7A was an endo-type alginate lyase, and its end-products were unsaturated oligosaccharides (degree of polymerization 2–6). Overall, due to the good characteristics, rTsAly7A can be used as a tool enzyme for producing AOS with high degree of polymerization. 

## Figures and Tables

**Figure 1 marinedrugs-21-00006-f001:**
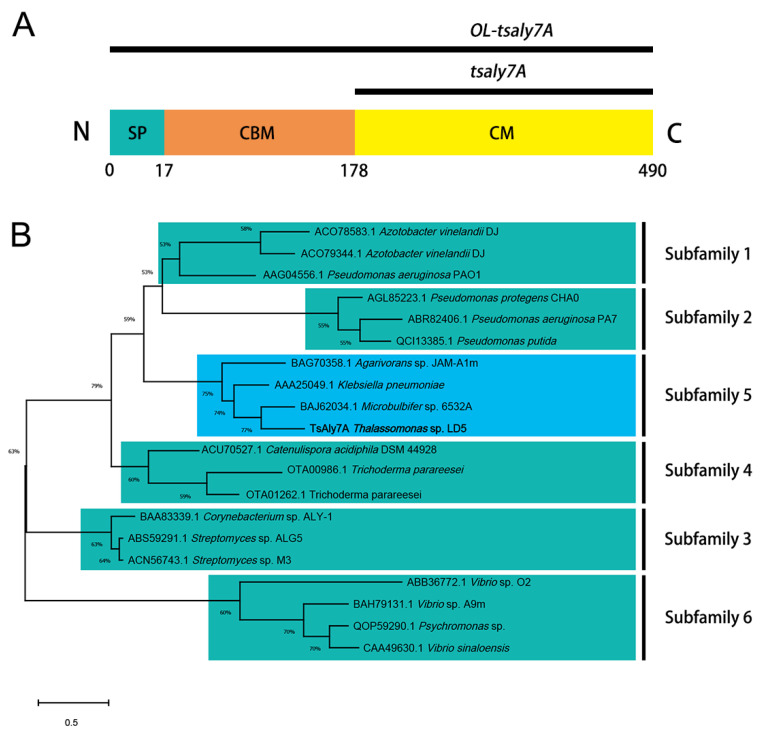
(**A**) Sequence analysis of TsAly7A. Domain structure of *OL-tsaly7A*. (**B**) Phylogenetic tree analysis of TsAly7A. The unrooted phylogenetic tree was constructed by the Maximum Likelihood method and JTT matrix-based model using MEGA X. Bootstrap analysis was computed with 1000 replicates, and bootstrap values below 50% were omitted. TsAly7A was marked with thickening in blue box. Subfamily 1, the first subfamily of PL7.

**Figure 2 marinedrugs-21-00006-f002:**
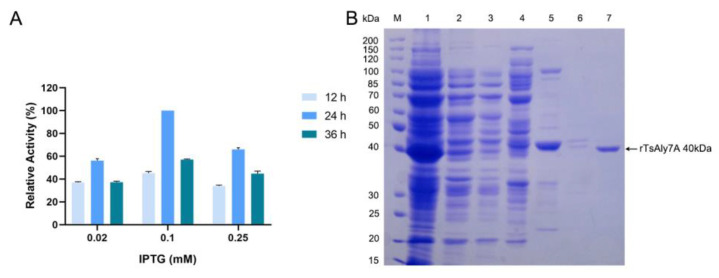
(**A**) Expression and purification of rTsAly7A. Relative enzyme activity of TsAly7A under different induction conditions. (**B**) SDS-PAGE of rTsAly7A. Lane M, protein standard marker; lane 1, crude enzyme; lane 2, flow-through; lane 3, elution by 0 mM imidazole; lane 4, elution by 25 mM imidazole; lane 5, elution by 75 mM imidazole; lane 6, elution by 150 mM imidazole; lane 7, elution by 300 mM imidazole, purified rTsAly7A.

**Figure 3 marinedrugs-21-00006-f003:**
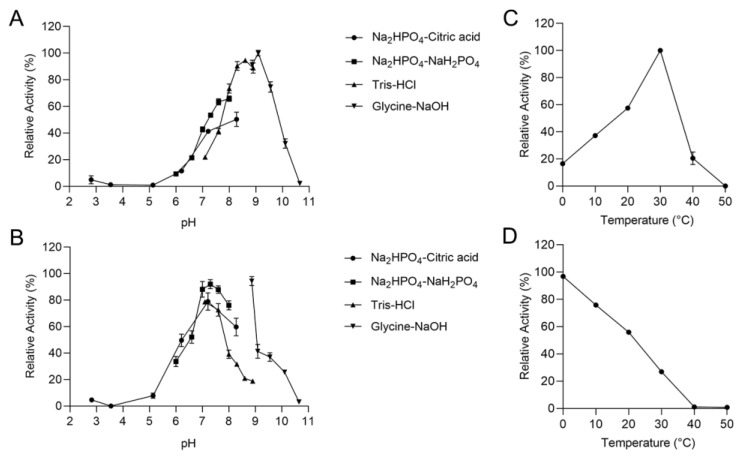
Biochemical properties of rTsAly7A. Optimal pH (**A**), pH stability (**B**), optimal temperature (**C**), thermal stability (**D**) of the rTsAly7A. The relative activity of 100% in (**A**,**C**) was determined at optimal condition. The original activity of 100% in (**B**,**D**) was determined before incubation at optimal condition.

**Figure 4 marinedrugs-21-00006-f004:**
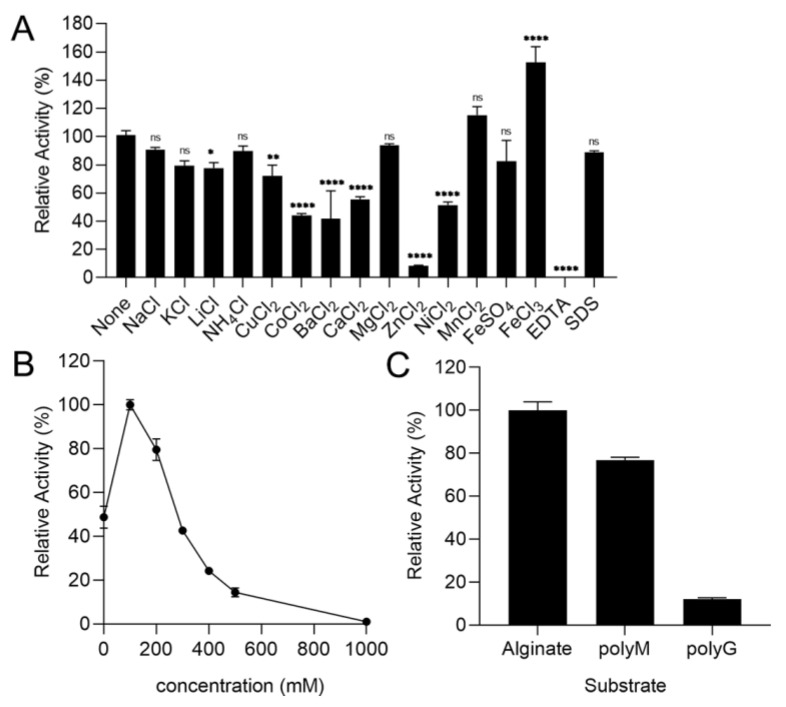
Effects of metal ions, chelator, and surfactant (1 mM) (**A**) and effects of NaCl concentrations (0–1 M) (**B**) on the activity of rTsAly7A. The substrate specificity of rTsAly7A (**C**). The relative activity of 100% was determined at optimal condition. **** for *p* < 0.0001, ** for *p* < 0.01, * for *p* < 0.05, ns for not significant.

**Figure 5 marinedrugs-21-00006-f005:**
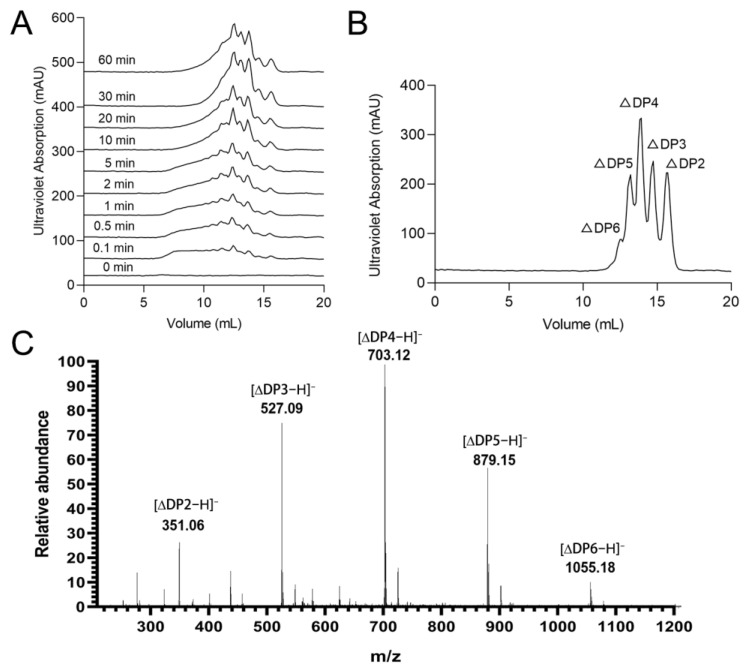
Degradation mode and end-products of rTsAly7A. The time-course of alginate degradation by rTsAly7A (**A**). SEC analysis of end-products of rTsAly7A (**B**). Mass spectra analysis of final product of rTsAly7A (**C**). ∆DP2, unsaturated alginate disaccharide.

## Data Availability

Not applicable.

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
