# Peer review of "Identification and Characterization of a New Cold-Adapted and Alkaline Alginate Lyase TsAly7A from Thalassomonas sp. LD5 Produces Alginate Oligosaccharides with High Degree of Polymerization"

_marinedrugs, 2022, doi:10.3390/md21010006_

Round 1

Reviewer 1 Report

Although this article is of interest as it deals with a cold-adapted and alkaline alginate lyase, the purpose of this study is not clearly stated in the introduction. Therefore, there is insufficient discussion.

I suggest that at least the following revises are necessary.

(1) I will suggest to rewrite “ nonetheless” into “whereas” at Lane 101.

(2) It is almost certain that rTsAly7A is a cold-adapted alginate lyase, but that doesn’t mean it should be used at room temperature at Lane 103-104. Because the enzyme activity is highly unstable at room temperature from the data of thermal stability.

(3) Why is the relative activity at 0 ℃ not 100 % ?

(4) It is necessary that each data in Figure 4 A requires statistical analysis, and reconsideration of the sentences at Lane 108-113, 176-180.

(5) The relative enzyme activity is slightly decreased with NaCl (Fig. 4A), while the data in figure 4B indicate that the relative activity is increased with NaCl. Is this not a contradiction ?

(6) I will suggest to delete “The enzyme activity ~ in Figure 4C.” at Lane 121-123.

(7) I will suggest to add the ambient temperature of collected the coastal sediments at Lane 157

(8) Authors need to make some evidences at Lane166-167. 

 That’s all.

Author Response

Response to Reviewer 1 Comments

Point 1: I will suggest to rewrite “nonetheless” into “whereas” at Lane 101.

Response 1: Thanks for your suggestion. As suggested by the reviewer, we have corrected the “nonetheless” into “whereas” at Lane 151.

Point 2: It is almost certain that rTsAly7A is a cold-adapted alginate lyase, but that doesn’t mean it should be used at room temperature at Lane 103-104. Because the enzyme activity is highly unstable at room temperature from the data of thermal stability.

Response 2: Thank you for pointing this out. According to the data of thermal stability, we found after incubation for an hour at 20 oC, it maintained half of the enzyme activity. So we suggested it can be used at room temperature.

Point 3: Why is the relative activity at 0 oC not 100 % ?

Response 3: Thank you for pointing this out. The 100% of the enzyme activity was measured before incubated and the enzyme activity at 0 oC was measured after incubated at 0 oC for one hour, so there may be decrease of enzyme activity. We should mention this in our article and we have added this point in Figure 3 at Lane 157-159.

Point 4: It is necessary that each data in Figure 4 A requires statistical analysis, and reconsideration of the sentences at Lane 108-113, 176-180.

Response 4: Thank you for your nice comments on our article. According to your suggestions, we have added significance analysis for each data as shown in Figure 4A. And then we have made reconsideration of the sentences at Lane 160-165, 251-255.

Point 5: The relative enzyme activity is slightly decreased with NaCl (Fig. 4A), while the data in figure 4B indicate that the relative activity is increased with NaCl. Is this not a contradiction ?

Response 5: Thank you for pointing this out. After we made significance analysis for each data in Figure 4A, we found there was no significance with 1 mM NaCl thanks to your nice suggestions.   So the concentration of NaCl used in effects of metal ions was 1 mM, which showed no significant effect, whereas the data in Figure 4B showed the relative activity is increased with 100 mM NaCl. So we should mention this in figure captions and we have added this point at Lane 177.

Point 6: I will suggest to delete “The enzyme activity ~ in Figure 4C.” at Lane 121-123.

Response 6: We think this is an excellent suggestion. Indeed, this sentence was unnecessary so it will be deleted in our article.

Point 7: I will suggest to add the ambient temperature of collected the coastal sediments at Lane 157.

Response 7: As suggested by the reviewer, we have looked up the temperature and salinity observation data in Chinese oceanic stations about XiaoMaiDao in National Marine Data Center, which showed that the coastal sediments had a temperature change between 5-25 oC all the year round. And we obtained Thalassomonas sp. LD5 in the coastal sediments with a temperature of 5 oC. So we changed our sentence at Lane 225.

Point 8: Authors need to make some evidences at Lane166-167.

Response 8: Thanks for your suggestion. As suggested by the reviewer, we have added more evidences to support the idea at Lane 233-234.

Reviewer 2 Report

The research presented by Yin et al. describe the activity of a new cold-adapted and alkline alginate lisase from Thalassomonas sp. to produce alginate oligosaccharides with a high degree of polymerization.

The introduction of the study is complete, but I miss a little bit more details about how AOS are useful, and how is working mechanism for example in the anti-obesity effects, moreover it would be interesting to explain in the introduction the different types of AOS depending on the bond type, ramifications, etc.

Different aspects should be modified before to be considered for publicacion:

Results Figure 1. The methods used for the phylogenetic analysis are not detailed, the algorithm used is not known, nor is it known which algorithm was used for the alignment. Such an analysis should be done using Maximum Likehood Parsimony instead of Neighbour Joining, which is much faster but less accurate for phylogenetic inferences. On the other hand, nodes with a boostrap value of less than 50 are unreliable and cannot be considered to be in distinct clades. This analysis should be reconsidered or clarified, both in results and methods.

Line 84 "data not showen" correct to "data not shown". Moreover, you have to show the results of the different tests to optimize the induction, at least in Supplementary materials.

Please rewrite the figure captions correctly, are poor in details and are not clear.

The total activity of the enzyme is not defined. All the activities showed are relative activities. But is impossible to compare with other lyases without more specific data such as total activity, Km or Vmax. The biochemical characterization is a little bit poor to describe a new enzyme.

Line 168 Vibrio sp. in italics please

Line 212 "MAGA-X" it is supposed to be "MEGA X"

Author Response

Response to Reviewer 2 Comments

Point 1: The introduction of the study is complete, but I miss a little bit more details about how AOS are useful, and how is working mechanism for example in the anti-obesity effects, moreover it would be interesting to explain in the introduction the different types of AOS depending on the bond type, ramifications, etc. 

Response 1: We feel great thanks for your professional review work on our article. According to your suggestions, we have supplemented several details at Lane 44-59.

Point 2: Results Figure 1. The methods used for the phylogenetic analysis are not detailed, the algorithm used is not known, nor is it known which algorithm was used for the alignment. Such an analysis should be done using Maximum Likelihood Parsimony instead of Neighbour Joining, which is much faster but less accurate for phylogenetic inferences. On the other hand, nodes with a boostrap value of less than 50 are unreliable and cannot be considered to be in distinct clades. This analysis should be reconsidered or clarified, both in results and methods. 

Response 2: According to your nice suggestions, we have made extensive corrections to our previous draft. We have detailed the phylogenetic analysis and have used Maximum Likelihood Parsimony to re-analysis our data at Lane 103-108.

Point 3: Line 84 "data not showen" correct to "data not shown". Moreover, you have to show the results of the different tests to optimize the induction, at least in Supplementary materials.

Response 3: We were really sorry for our careless mistakes. Thank you for your reminder. As for the results of the different tests to optimize the induction, we added the data in Figure 2A at Lane 117-122.

Point 4: Please rewrite the figure captions correctly, are poor in details and are not clear.

Response 4: Thank you again for your positive comments and valuable suggestions to improve the quality of our manuscript. We have rewrite the figure captions as Lane 103-108, 117-122, 155-159, 176-180, 212-215. 

Point 5: The total activity of the enzyme is not defined. All the activities showed are relative activities. But is impossible to compare with other lyases without more specific data such as total activity, Km or Vmax. The biochemical characterization is a little bit poor to describe a new enzyme.

Response 5: Thank you for pointing this out. We agree that this is an important consideration, we mentioned that the specific activity of rTsAly7A was 1536.36 U/mg at Lane 107. And as we knew, TsAly7A was one good alginate lyase compared with other lyases.  

Point 6: Line 168 Vibrio sp. in italics please

Response 6: We are really sorry for our careless mistake and we have corrected this mistake. Thank you for your reminder. 

Point 7: Line 212 "MEGA-X" it is supposed to be "MEGA X"

Response 7: Thank you for pointing this out. "MEGA-X" has been corrected to "MEGA X".

Round 2

Reviewer 2 Report

Thanks for your corrections, the manuscript now is improved. :)